# Vaccination by Two DerG LEAPS Conjugates Incorporating Distinct Proteoglycan (PG, Aggrecan) Epitopes Provides Therapy by Different Immune Mechanisms in a Mouse Model of Rheumatoid Arthritis

**DOI:** 10.3390/vaccines9050448

**Published:** 2021-05-02

**Authors:** Daniel H. Zimmerman, Katalin Mikecz, Adrienn Markovics, Roy E. Carambula, Jason C. Ciemielewski, Daniel M. Toth, Tibor T. Glant, Ken S. Rosenthal

**Affiliations:** 1CEL-SCI Corporation, 8229 Boone Blvd, Vienna, VA 22182, USA; RCarambula@cel-sci.com (R.E.C.); JCiemielewski@cel-sci.com (J.C.C.); 2Department of Orthopedic Surgery, Rush University Medical Center, 1735 W. Harrison St, Chicago, IL 60612, USA; Katalin_Mikecz@rush.edu (K.M.); Adrienn_Markovics@rush.edu (A.M.); Daniel.Toth@mssm.edu (D.M.T.); Tibor_Glant@rush.edu (T.T.G.); 3Division of Rheumatology, Department of Medicine, Icahn School of Medicine at Mount Sinai, 1468 Madison Ave, New York, NY 10029, USA; 4Department of Basic Medical Sciences, Augusta University/University of Georgia Medical Partnership, 1425 Prince Avenue, Athens, GA 30602, USA; kenneth.rosenthal@uga.edu

**Keywords:** peptide vaccine, immunotherapy, rheumatoid arthritis, Proteoglycan (PG, aggrecan), PG G1 domain-induced arthritis

## Abstract

Rheumatoid arthritis (RA) can be initiated and driven by immune responses to multiple antigenic epitopes including those in cartilage proteoglycan (PG, aggrecan) and type II collagen. RA is driven by T helper 1 (Th1) or Th17 pro-inflammatory T cell responses. LEAPS (Ligand Epitope Antigen Presentation System) DerG peptide conjugate vaccines were prepared using epitopes from PG that elicit immune responses in RA patients: epitope PG70 (DerG-PG70, also designated CEL-4000) and the citrullinated form of another epitope (PG275Cit). The LEAPS peptides were administered alone or together in Seppic ISA51vg adjuvant to mice with PG G1 domain-induced arthritis (GIA), a mouse model of RA. Each of these LEAPS peptides and the combination modulated the inflammatory response and stopped the progression of arthritis in the GIA mouse model. Despite having a therapeutic effect, the DerG-PG275Cit vaccine did not elicit significant antibody responses, whereas DerG-PG70 (alone or with DerG-PG275Cit) induced both therapy and antibodies. Spleen T cells from GIA mice, vaccinated with the DerG LEAPS peptides, preferentially produced anti-inflammatory (IL-4 and IL-10) rather than pro-inflammatory (IFN-γ or IL-17) cytokines in culture. Similarly, cytokines secreted by CD4+ cells of unvaccinated GIA mice, differentiated in vitro to Th2 cells and treated with either or both DerG vaccine peptides, exhibited an anti-inflammatory (IL-4, IL-10) profile. These results suggest that the two peptides elicit different therapeutic immune responses by the immunomodulation of disease-promoting pro-inflammatory responses and that the combination of the two LEAPS conjugates may provide broader epitope coverage and, in some cases, greater efficacy than either conjugate alone.

## 1. Introduction

Rheumatoid arthritis (RA) is an autoimmune disorder characterized by systemic inflammation, progressive joint deterioration, and loss of mobility. According to the CDC, an estimated 1.5 million people in the USA had RA in 2007, with approximately 50 million worldwide having some form of arthritis [1]. Another more recent report looking at more selective populations in 2019 stated 4,600,000 cases in 2019 [2]. In 2012, three of the ten best-selling drugs in the USA were RA biologics that treated arthritis symptoms by ablation of activated immune cells or neutralization of pro-inflammatory cytokines.

Autoimmune diseases can be initiated and driven by cytokines produced by T cells, B cells, macrophages, dendritic cells (DCs) and other cells in response to multiple antigenic epitopes and pro-inflammatory stimuli. In RA, some of the prominent epitopes driving disease are from cartilage macromolecules such as proteoglycan (PG, aggrecan) and type II collagen [3,4]. Pro-inflammatory cytokines that promote or are secreted by T helper 1 (Th1) cells (IL-12, IFN-γ, TNF-α, IL-2) or Th17 cells (IL-23, IL-17, TNF-α, IL-22) often drive RA disease whereas cytokines secreted by Th2 cells (IL-4, IL-5, IL-10) and regulatory T (Treg) cells (TGF-β and IL-10) are anti-inflammatory and regulatory [4].

Immunomodulation of the disease promoting pro-inflammatory responses is a potential alternative to current therapies for RA [5,6,7]. Rosenthal et al. [3] reviewed and discussed other therapies for RA in relation to LEAPS conjugates and the differences in mechanism of action and targets. A sampling of more recent papers within the last few years of research and reviews of proposed or experimental modern therapies using cytokines targeted via soluble receptors or monoclonal antibodies for IL-6, IL-17 and Janus kinase inhibitors show promises but still limitations of unidirectional and all downward [8,9,10,11,12,13,14,15]. We have developed a peptide vaccine technology that can be protective or therapeutic, which we refer to as Ligand Epitope Antigen Presentation System (LEAPS). Attachment of an Immune Cell Binding Ligand (ICBL) peptide to a T cell epitope-containing peptide promotes immunogenicity and determines the resultant response [16]. LEAPS technology products are administered subcutaneously with adjuvants (such as Seppic ISA51vg) or via injection of peptide vaccine-loaded DCs [17,18].

Conjugation of the J ICBL (DLLKNGERIEKVE), from human beta-2-microglobulin [19], to disease-related peptides promotes Th1 responses by activating DC precursors to mature and produce IL-12p70 [17,18]. J-LEAPS conjugate vaccines or J-LEAPS induced DCs have been shown to be protective in animal models of viral infections such as herpes simplex virus type 1 (HSV-1) and influenza A [17,18] and therapeutic for inflammatory diseases in models such as the Th17 driven experimental autoimmune myocarditis (EAM) and collagen induced arthritis (CIA) [16,20]. Conjugation of the DerG peptide ICBL (DGQEEKAGVVSTGLI), from the beta chain of human major histocompatibility complex II, to disease related peptides (depicted as a conceptual diagram in Figure 1) promotes immune responses with Th2 associated cytokine profiles [21]. A DerG-LEAPS peptide was therapeutic for the proteoglycan (PG)-induced arthritis (PGIA) and PG G1 domain induced arthritis (GIA) mouse models of RA [21], but DerG-LEAPS vaccines are not protective or therapeutic in the HSV-1, EAM or CIA models [16,17,20].

Animal models of autoimmune conditions such as multiple sclerosis, uveitis and RA [22] are driven by pro-inflammatory antigen-specific Th1 or Th17 responses [22,23,24,25] depending upon the disease driving antigen and method of induction as well as animal type, strain, and age [3,22]. For example, CIA is induced in young male DBA/1 mice and is characterized by a predominant Th17 cytokine response (IL-17, IL-1β, TNF-α) [26]. In the CIA model, a J-LEAPS conjugate incorporating a collagen epitope (referred to as CEL-2000) was therapeutic via the up-regulation of IL-12 and IFN-γ activity and down-modulation of IL-17, IL-1β and TNF-α [16]. The PGIA and GIA models of RA are induced by immunization of aging female BALB/c mice with human cartilage PG and/or the recombinant first G1 domain (rhG1) of human PG, respectively [27,28]. These animal models resemble human disease with the presence of rheumatoid factor as well as anti-citrullinated protein antibodies in serum [27,28]. PGIA and GIA are characterized by predominant Th1 type (IFN-γ) cytokine production. A DerG-LEAPS conjugate of the PG70 epitope peptide (referred to as CEL-4000) was therapeutic in these models via up-modulation of Th2 and regulatory responses [21].

By acting on T cell subsets driving disease in an antigen-specific manner, LEAPS immunotherapies relieve inflammatory symptoms and modulate, rather than ablate, ongoing disease-promoting cytokine and cellular responses in models of autoimmune inflammatory diseases [3,33]. The J-LEAPS immunotherapies down-modulate Th17 responses and the DerG-LEAPS vaccines down-modulate Th1 responses, both of which can play a role in RA for humans and in animal models of the disease [4,21,22]. As such, determination of the T cell response driving an individual’s RA disease would allow a more personalized choice of the LEAPS vaccine for treatment [3,33]. 

In this study, the GIA animal model of RA was utilized to test a new DerG conjugate incorporating the citrullinated form of the PG275 (PG275Cit) peptide administered alone and in combination with DerG-PG70 (CEL-4000). T cell and antibody responses to both of these peptides have been detected in mice with PGIA or GIA and in RA patients [34,35]. Citrullination is a common post-translational modification found in numerous RA antigens including PG [35,36]. The conversion of arginine (R) to citrulline (Cit) is catalyzed by peptidyl-arginine deiminase enzymes [37,38], some of which are also associated with RA [39,40,41]. Most RA patients produce anti-citrullinated protein/peptide antibodies [22,42]. 

Each of the DerG-LEAPS vaccine formulations were evaluated separately and in combination for therapeutic intervention of ongoing disease, antibody production, and effects on relevant immune responses in the GIA model of RA. The ability of the vaccines to enhance specific Th cell types and the predilection towards anti-inflammatory phenotypes was examined using in vitro Th differentiation and assaying the cytokines elicited by the vaccine peptides.

## 2. Materials and Methods

### 2.1. Peptides and Antigen

LEAPS peptides used for vaccine preparation were supplied with free amino terminus and amidated C-terminus as lyophilized acetate salts at ≥90% purity by RP-HPLC and MS by 21st Century (Marlborough, MA, USA), Ambiopharma (North Augusta, SC, USA), or Bachem (Torreance, CA, USA). Peptides used for serum anti-peptide antibody detection were supplied by Biomatik (Wilmington, DE, USA) in biotinylated form. These peptides have the following amino acid (aa) sequences: PG70 (ATEGRVRVNSAYQDK), PG275Cit (MDMCSAGWLAD{Cit}SVR), DerG (DGQEEKAGVVSTGLIGGG), J (DLLKNGERIEKVEGGG), OVA (chicken egg ovalbumin, AAHAEIMEAGREVVG), DerG-PG70 (see aa sequences above), DerG-PG275Cit (see aa sequences above) and DerG-M2e (DGQEEKAGVVSTGLIGGGSLLTEVETPIRNEWGSRSNDSSD) [18]. All biotinylated peptides were extended with a GGG sequence at the N terminus to provide a spacer between the peptide and the N-terminally attached biotin tag. The recombinant human PG G1 domain (rhG1) antigen used for GIA induction in the mice and in vitro antigen-restimulation of GIA spleen cells was prepared as described previously [21,27]. 

### 2.2. Mice, Disease Induction and Assessment of GIA

Retired breeder female BALB/c mice were purchased from the National Cancer Institute (NCI) colony of Charles River (Wilmington, MA, USA). GIA was induced by 3 intraperitoneal injections of 40 µg rhG1 in dimethyl-dioctadecyl ammonium bromide (DDA) adjuvant 3 weeks apart as described [27,28]. After the third injection, the mice were inspected daily (For related information see the caption of Appendix A) for arthritis symptoms. Swelling and redness of the paws were visually scored. Arthritis scores ranged from 0 to 4 per paw, resulting in a cumulative score of 0–16 per mouse. Mice were sorted into treatment groups, each group having a similar mean arthritis score of ~2.5 ± 0.5 on day 0 of LEAPS therapy (i.e., just before the first vaccination).

### 2.3. Preparation and Administration of Vaccines

The vaccines were prepared by dissolving the LEAPS peptides in sterile phosphate-buffered saline (PBS; pH 7.4) at 100 nmoles of each peptide per 100 µL PBS and emulsifying the peptide solutions with the same volume of Seppic’s Montanide ISA51vg adjuvant (Seppic Inc., Fairfield, NJ, USA) per injection per mouse. PBS emulsified with adjuvant served as a peptide-free adjuvant control. The vaccines were administered subcutaneously on therapy days 0 and 14.

Arthritis scores were determined 3 times per week between days 0 and 35 after the first vaccination as described previously [21]. All experiments involving animals were approved by the Institutional Animal Care and Use Committee of Rush University Medical Center (Chicago, IL, USA). All animal experiments complied with the NIH and NRC guides for the care and use of laboratory animals [43].

### 2.4. Collection of Specimens from Vaccinated Mice

Animals were euthanized by CO_2_ inhalation on day 35 after the first LEAPS peptide or adjuvant control vaccination. Blood was collected prior to euthanasia via facial vein puncture with sterile lancets. Serum from each mouse was obtained from the blood samples and stored at −70 °C until use. The spleens were removed aseptically from the euthanized animals and processed for cell culture. Hind limbs were dissected and fixed in 10% buffered formalin for histopathology.

### 2.5. Histopathology

The hind limbs of vaccinated GIA and naïve (negative control) mice were decalcified, embedded in paraffin, and sectioned. Adjacent tissue sections were stained separately with toluidine blue (TB) or hematoxylin and eosin (H&E). Microscopic analysis and scoring of joint damage were carried out under code by independent investigators at Bolder BioPath (Boulder, CO, USA). The individual parameters (each scored from 0 to 5) were inflammation, pannus formation, cartilage damage, bone resorption, and periosteal bone formation. Representative joint tissue photographs were taken from each group of mice. A summed histopathology score (ranging from 0 to 25) was also calculated for each mouse, as described previously [21].

### 2.6. Measurement of Serum Anti-Peptide Antibody Levels

Serum levels of IgG antibodies to LEAPS and control peptides of vaccinated mice were determined by ELISA as described [35]. In brief, biotinylated peptides (DerG-PG70, DerG-PG275Cit, PG70, PG275Cit, DerG, J, or OVA) were immobilized on Neutravidin-coated and pre-blocked plates (Pierce, Rockford, IL, USA) at 2.5 µg/well in 100 µL Tris-buffered saline. After washing, mouse serum samples at 1:1000 dilution were incubated with the immobilized peptides for 1 h at room temperature. Peptide-bound IgG antibodies were detected by incubation with horseradish peroxidase-labeled anti-mouse IgG (BD Biosciences, San Diego, CA, USA) followed by addition of 3,3′,5,5′-tetramethylbenzidine substrate (BD Biosciences) for color development. Optical density (OD) values at 450 nm were read by an ELISA reader (BioTek ELX808, Winooski, VT, USA) [21].

### 2.7. Spleen Cell Cultures

Spleens, harvested from euthanized GIA or naïve mice, were disrupted by extrusion through sterile metal screens. Red blood cells were eliminated by lysis in ice-cold ammonium chloride-containing hemolysis buffer. Single cells were suspended in Dulbecco’s Modified Eagle Medium (DMEM; Sigma-Aldrich, St. Louis, MO, USA) containing 50 µM 2-mercaptoethanol, antibiotics and antimycotics (Sigma-Aldrich) and 10% fetal bovine serum (FBS; Hyclone, Logan, UT, USA).

The spleen cells from the in vivo rhG1-immunized and vaccinated GIA mice were cultured in the presence of rhG1 antigen (in vitro restimulation with 7.5 µg/mL) for 4 days before harvest for flow cytometry and cytokine secretion assays. For comparison with the antigen-restimulated cells, non-restimulated (no rhG1) cells were used as baseline controls in the cytokine assays.

In separate experiments, CD4+ spleen cells from unvaccinated GIA mice were used to examine the effects of CEL-4000 on cytokine production by in vitro differentiated Th1, Th2, or Th17 subsets. CD4+ cells were isolated from the spleens of GIA mice by negative immunomagnetic selection using the EasySep CD4+ cell separation kit (StemCell Technologies, Vancouver, BC, Canada). Antigen-presenting cells (APCs) were prepared from separate aliquots of GIA spleen cells by depleting T cells via positive immunomagnetic depletion (StemCell Technologies) with a biotinylated anti-CD3 antibody (Biolegend, San Diego, CA, USA). The CellXVivo Th1, Th2, or Th17 mouse differentiation kits (R&D Systems, Minneapolis, MN, USA) were used to induce differentiation into the various subsets of Th CD4+ T cells, according to the manufacturer’s recommendations and as reported by other groups [44,45]. CD4+ cells 1.5 × 10^6^/mL were mixed with the Th1, Th2, or Th17 differentiation reagents, 0.5 × 10^6^/mL APCs (in a total volume of 0.6 mL) and cultured for 5 (Th17) or 6 (Th1 and Th2) days in the presence of rhG1 (7.5 µg/mL) and in the absence (None) or presence of CEL-4000 (15 µM). Undifferentiated Th0 CD4+ cells were cultured for the same time without the differentiation reagents and used as reference controls. In a follow-up study, the Th2 differentiation condition was used to test the effects of additional peptides on cytokine secretion. In brief, spleen CD4+ cells and APCs were prepared from unvaccinated GIA mice as described above. The cells were mixed with the Th2 differentiation reagents and cultured for 6 days in the presence of rhG1 and in the absence (None) or presence of one of the following peptides: CEL-4000, Derg-PG275Cit, PG70, DerG, DerG-M2e, J-PG70, or the combination of CEL-4000 and DerG-PG275Cit (15 µM each). The differentiation reagents were washed out after the differentiation period, and the cells were cultured for 2 additional days in the presence of rhG1 and in the absence or presence of the above-listed peptides. Culture supernatants were collected from each well for measurement of secreted cytokines.

To test the ability of the CEL-4000 LEAPS peptide to alter the differentiation of polyclonally activated naïve CD4+ cells, we separated the CD4+ population and APCs from the spleens of naïve (non-immunized, unvaccinated) mice using immunomagnetic selection as described above. The CD4+ cells and APCs were mixed with the Th1, Th2, or Th17 differentiation kit reagents. Unlike the GIA cells, the naïve T cells were activated polyclonally with anti-CD3/CD28 antibodies (included in the kits) in the absence or presence of the CEL-4000 peptide (15 µM). On day 5 or 6 of culture, the differentiation reagents were washed out and the cells were re-stimulated with anti-CD3/CD28 antibodies for 2 days before the harvest of supernatants for secreted cytokine assays. CEL-4000 (15 µM) was present (or absent from control wells) during the entire course of cell culture.

### 2.8. Flow Cytometry

The cell phenotypes of spleen cells from the vaccinated GIA mice were determined by expression of cell-surface CD4 and intracellular (IC) staining with antibodies to cytokines (IFN-γ for Th1 cells, IL-4 and IL-10 for Th2 cells, IL-17A for Th17 cells) as well as to FoxP3 for Treg cells [46,47]. Cell culture, immunostaining and flow cytometry were performed as described previously [21,42]. To determine the effects of vaccination on the cells, cytokine expression was represented by multiplying mean fluorescence intensity (MFI) by the % of cytokine positive CD4+ cells. Final results were presented as the ratios of key anti-inflammatory (IL-4, IL-10) to pro-inflammatory (IFN-γ, IL-17A) cytokine expression.

### 2.9. Measurement of Cytokines (IL-1β, IL-2, IL-4, IL-6, IL-10, IL-17A, IFN-γ and TNF-α) Secreted by Spleen Cells

Cytokines secreted by the above-described cells into the culture media were assayed by the MagPix method using Multiplex (9-plex, 8-plex or 6-plex) mouse cytokine kits purchased from R&D Systems. The data shown in the figures were expressed as the concentrations of key (IL-4, IL-10, IFN-γ, and IL-17) cytokines secreted, as determined by the manufacturer’s standards and software package as pg/mL or as the ratios of anti-inflammatory (IL-4 or IL-10) to pro-inflammatory (IFN-γ or IL-17A) cytokines, as described previously [21] by dividing the pg/mL or one cytokine by the concentration of the other cytokine (also in pg/mL) for the same four cytokines evaluated in Section 2.8. We provide entire sets of graphs of cytokine concentrations in Appendix A as Appendix A. We did not determine the ratios for IL-6 and TNF-α to either IL-4 or IL-10 as both IL-6 and TNF-α concentrations were very low and would have distorted the ratios in comparison to the other 4 cytokine sets. Furthermore, the molecular weights or masses were not supplied to make the conversion to molarity, and the probability of differential glycosylation of live cell-produced cytokines would have further complicated the issue.

### 2.10. Experimental Study Design, Animal Use, Powering and Statistical Analysis of Data

We followed the ARRIVE guidelines (For animal research: reporting in vivo experiments) in study design, powering of assays, animal use, selection criteria, blinding, and transparency [48,49]. Data are expressed as mean ± SEM unless indicated otherwise. Statistical analysis was performed using GraphPad Prism 7.04 software package (GraphPad, La Jolla, CA, USA). Results were analyzed using one- or two-way analysis of variance (ANOVA) followed by multiple comparison tests as indicated in figure legends for data from multiple groups. *p* values less than 0.05 were accepted as statistically significant.

## 3. Results

### 3.1. LEAPS Vaccines Limit Arthritis Progression

For an introductory study (study 1), mice presenting with early-phase GIA disease (*n* = 8 per group) were treated with adjuvant only (PBS in adjuvant control) or the following LEAPS vaccine peptides in adjuvant: CEL-4000, DerG-PG275Cit, or the combination of CEL-4000 and DerG-PG275Cit. The severity of disease in the adjuvant control and LEAPS vaccinated groups, as shown in Figure 2, was assigned numerical values as arthritis scores. After the first treatments with CEL-4000 or DerG-PG275Cit, there was an initial reactive stage at day 4, followed approximately one week later by noticeably higher arthritis scores for the adjuvant only control group compared to the LEAPS peptide treated groups. For all treatments, arthritis scores peaked on day 14, but these values were still much lower for the LEAPS vaccinated mice than those in the control group. The scores for the LEAPS treated groups then dropped quickly after a booster LEAPS vaccination treatment on day 14, and by day 35, all scores were below those in the adjuvant treated control group. *p* values of statistically significant differences between the arthritis scores of vaccinated GIA mice during the course of arthritis are shown in Appendix A. These results indicate therapeutic actions for CEL-4000, DerG-PG275Cit, and the combination of these peptides over the course of vaccine treatment.

Figure 3 shows TB and H&E-stained sections of joint tissues taken on day 35 from representative mice of each of the four treatment groups described above. Adjuvant treated control animals displayed marked inflammation and moderate cartilage damage with some pannus formation and bone resorption as well as severe periosteal bone formation in the ankle and several digit joints. CEL-4000 treated animals displayed only moderate inflammation (see H&E staining) and only mild cartilage damage (see TB staining) in the ankle and a single digit joint. Mice treated with the combination of CEL-4000 and DerG-PG275Cit displayed only mild inflammation and cartilage damage with minimal pannus formation and bone resorption as well as moderate periosteal bone formation, but only in the ankle. DerG-PG275Cit treated animals displayed moderate inflammation and mild cartilage damage in the ankle and several digit joints. In summary, tissues obtained from LEAPS vaccinated mice demonstrated less joint pathology than the control animals, which was consistent with the arthritis scores (see Figure 2).

Treatment of GIA mice with the same vaccines was repeated in a confirmatory study (study 2). As shown in Figure 4, although the time course of the disease was different from that observed in study 1, all of the LEAPS peptide-vaccinated mice had lower arthritis scores than the controls from day 14 to day 35 (see study statistics in Appendix A) indicative of a block in disease progression.

In sum, the course of arthritis was different from the pattern seen in study 1 (see Figure 2), but the outcomes of both studies were similar. Detailed histopathological analysis of the joints (Figure 5) indicated that all parameters of joint damage were reduced including inflammation, pannus formation, cartilage destruction, bone resorption, and periosteal bone formation in mice treated with CEL-4000 compared to control adjuvant treated mice. The joints of mice treated with a combination of CEL-4000 and DerG-PG725Cit vaccine also showed a trend of reduced destruction (Figure 5A and Appendix A). However, in the summed histopathology scores (Figure 5B), statistically significant reduction in overall tissue damage was found only in the CEL-4000 treated group in comparison with the other treatment groups. Representative TB-stained joint sections from this study are shown in Appendix A.

### 3.2. LEAPS Vaccines Elicit Different Serological Responses

IgG antibodies in serum obtained from mice from study 2 were compared by ELISA for reactivity to each of the LEAPS vaccines or LEAPS component peptides using ovalbumin (OVA) as an irrelevant control peptide. As shown in Figure 6, CEL-4000 treatment of GIA mice induced antibodies to both the DerG and PG70 elements. DerG-PG275Cit elicited low or background levels of antibodies to its cognate peptides. Mice receiving the combination of CEL-4000 and DerG-PG275Cit produced antibodies that resembled the response to CEL-4000 alone (Figure 6). These results indicate a difference in the humoral arm of the in vivo immune responses to the DerG-PG275Cit and CEL-4000 vaccines.

### 3.3. Intracellular Cytokine Content and FoxP3 in Spleen Cells of Vaccinated GIA Mice

Intracellular (IC) cytokine and FoxP3 expression of CD4+ spleen cells obtained from the vaccinated GIA mice from study 2 were evaluated by flow cytometry. For IC cytokines and FoxP3, the expression results were calculated as described in the Methods (MFI × % of positive cells). The data are presented in Figure 7A as a ratio of the IC expression of anti-inflammatory cytokines (IL-4 or IL-10) to the expression of pro-inflammatory cytokines (IFN-γ or IL-17A) (MFI and % values for each cytokine are available from the authors upon request). The expression ratios for cells from CEL-4000 vaccinated mice had significantly elevated IL-10: IFN-γ ratios with a trend towards elevated anti-inflammatory cytokine ratios. Cells from DerG-PG275 vaccinated animals had significantly elevated IL-4: IFN-γ and also IL-10:IFN-γ ratios (Figure 7A). A trend towards anti-inflammatory cytokine dominance was also observed for the combination treatment. However, there were no statistically significant differences in FoxP3 expression between the control and LEAPS treated groups (Figure 7B).

### 3.4. Soluble Cytokine Production by Spleen Cells from Control and LEAPS Peptide Vaccinated GIA Mice

We examined cytokine secretion by in vitro antigen-restimulated or non-restimulated spleen cells. The data for all cytokines evaluable are shown in Appendix A and includes IL-6 and TNF-α not used in the determination of ratios as explained earlier in the materials and methods 2.9 for several reasons. The cells were harvested on day 35 from each of the adjuvant-treated and LEAPS vaccinated GIA mice from study 2 and cultured for 4 days in the absence (non-restimulated, Figure 8A) or in the presence (restimulated) of rhG1 antigen (Figure 8B), and then the cytokines released into the medium were assayed. Results are shown as ratios of anti-inflammatory IL-4 or IL-10 to pro-inflammatory IFN-γ or IL-17A cytokine concentrations. As shown in Figure 8A, spleen cells from LEAPS vaccinated GIA mice all produced higher ratios of anti-inflammatory to pro-inflammatory cytokines than the cells from adjuvant-treated control animals, especially with regard to the IL-4 or IL-10 to IFN-γ ratios. Similar results were obtained for spleen cells restimulated in vitro with rhG1 antigen, although the anti-inflammatory to pro-inflammatory cytokine ratios were generally lower (Figure 8B) than in the case of non-restimulated cells. Ratios with IL-17 followed the trend but were not statistically significant in this Th1 driven animal model. Additional cytokine results for IL-1β, IL-2, IL-6, and TNF-α were also collected, but were unremarkable (data available from repository materials Appendix A).

### 3.5. Cytokine Responses of In Vitro Th Differentiated Spleen CD4+ Cell Subsets from Unvaccinated GIA Mice or from Naïve Mice

We sought to determine the effects of CEL-4000 on CD4+ spleen cells during in vitro differentiation into Th1, Th2, or Th17 subsets. First, we tested CEL-4000 using CD4+ cells and APCs from naïve BALB/c mice. The differentiation conditions were set up using CellXVivo kit reagents as described in the Methods, and undifferentiated (Th0) cells were used as reference controls. The naïve CD4+ cells were stimulated polyclonally with anti-CD3 and anti-CD28 antibodies and cultured in the absence (None) or presence of CEL-4000 peptide. Although polyclonal activation of naïve CD4+ cells elicited robust cytokine production, CEL-4000 had no significant effect on the cytokines secreted by Th0 cells or during differentiation into Th1, Th2, or Th17 subsets (Appendix A).

Since the CEL-4000 vaccine showed high in vivo therapeutic efficacy when injected into GIA mice (see Figure 1, Figure 2, Figure 3 and Figure 4), we next asked whether the CEL-4000 peptide could influence the polarization of Th cell subsets of unvaccinated GIA mice during in vitro differentiation. CD4+ T cells and APCs were isolated from the spleens of unvaccinated GIA mice, and restimulated with rhG1 under Th1, Th2, or Th17 differentiation conditions. All cells were cultured in the absence (None) or the presence of CEL-4000 as described in the Methods. CEL-4000 enhanced the production of anti-inflammatory cytokines from Th2 differentiated cells from GIA mice, but had no influence on Th0 cells or cells cultured under Th1 or Th17 differentiation conditions (Appendix A). As in the case of cytokine responses following in vivo CEL-4000 vaccination of GIA mice (see Figure 7), the in vitro differentiation study confirmed that the LEAPS peptides modulated the immune responses by upregulating anti-inflammatory cytokine production by Th2 differentiated cells originally harvested from unvaccinated GIA diseased mice.

To gain insight into the in vitro responses of Th2 cells to other peptides (vaccine components or control peptides), we tested cytokine secretion in the absence (None) or presence of CEL-4000 or other peptides, including peptides used to make up the conjugates (DerG, PG70), the J ICBL version of CEL-4000 (J-PG70) and an irrelevant DerG LEAPS conjugate (DerG-M2e). As seen in Figure 9 for Th2 differentiated GIA cells, the DerG and PG70 containing peptides elicited elevated ratios of anti-inflammatory to pro-inflammatory cytokines with CEL-4000 and DerG-PG275Cit being the most effective inducers of the anti-inflammatory cytokine response. In contrast, the anti-inflammatory to pro-inflammatory cytokine ratios for J-PG70 were orders of magnitude lower. These results are consistent with the Th2 promoting action of DerG and the predilection of J-PG70 to induce IFN-γ (and IL-12) associated Th1 responses [17]. Additional cytokine results for IL-1β, IL-2, IL-6, and TNF-α were also collected, but were unremarkable (data available from the authors upon request).

## 4. Discussion

In this study, we introduced a vaccine containing a PG epitope in citrullinated form, DerG-PG275Cit, which was shown to be immunogenic in the mouse model and in patients with RA [35]. We demonstrated that vaccination with DerG-PG70 (CEL-4000), DerG-PG275Cit and the combination of these LEAPS peptides provided immunomodulatory therapy for mice with GIA RA-like symptoms. The therapy was elicited in GIA, a mouse model that resembles the human disease [21], and the treatments were provided after disease was initiated as would be the case for patients with RA. These studies extend those described in [21] to include the second peptide vaccine and also to examine the nature of the immunomodulation of both peptides. No previous studies were published about the effect of LEAPS peptides on in vitro differentiated cells from Th0 to Th1, Th2, Th17 or Treg cells. We further showed that educated (antigen-experienced) spleen cells from diseased animals were essential; naïve spleen cells even if differentiated in vitro and treated with the antigen were not affected by the DerG-PG70 conjugate. Furthermore, Markovics et al. [35] used a PG275Cit peptide, but did not examine any LEAPS conjugate.

The vaccines immunomodulated the ongoing pro-inflammatory responses, as indicated by increased ratios of secreted anti-inflammatory to pro-inflammatory cytokines in the spleen cell cultures of the LEAPS treated mice, consistent with our earlier observations with CEL-4000 for spleen cell and serum cytokines [21]. Similarly, the ratios of IC cytokines within spleen cells of LEAPS vaccinated GIA mice indicated that CD4+ cells contained more anti-inflammatory than pro-inflammatory cytokines. Demonstration of the therapeutic efficacy following treatment with the combination of CEL-4000 and DerG-PG275Cit peptides shows that they do not interfere with each other.

CEL-4000 and DerG-PG275Cit both appear to act by up-modulation of Th2 responses in vivo. This is indicated by increased ratios of the anti-inflammatory cytokines IL-4 and IL-10 (produced mainly by Th2 cells) to pro-inflammatory IFN-γ (Th1-associated) or IL-17 (Th17-associated) cytokines [50], which were detected in the supernatants of spleen cell cultures and inside the CD4+ population collected from the vaccine treated GIA mice. The lack of a difference in FoxP3 expression within spleen cells from LEAPS vaccine-treated compared to control diseased mice suggests that classical Treg cells are not activated by these vaccines. However, the contribution of unconventional Treg cells to the anti-inflammatory cytokine profile of the spleen cells from vaccinated GIA mice cannot be excluded. For example, anti-inflammatory/regulatory IL-10 [51] is produced by Th2 and other cells including inducible type 1 regulatory T cells (Tr1) [52]. Tr1 cells are induced-regulatory T cells that do not express FoxP3 and can arise from naïve T cells, Th1, Th2, or Th17 cells in an IL-10-rich environment. Although not specifically tested in this study, some of the IL-10 may have come from Tr1 cells.

Even though both CEL-4000 and DerG-PG275Cit LEAPS conjugates were highly effective therapeutics in the GIA model of RA and both elicited favorable anti-inflammatory T cell cytokine responses, the vaccines appear to act by different mechanisms. This is indicated by differences in anti-peptide antibody production by the vaccinated mice. CEL-4000 induced serum antibodies which recognized DerG and PG70 and their conjugates and the DerG component even within the DerG-PG275Cit peptide. In contrast, the DerG-PG275Cit vaccine did not induce remarkable antibody responses to its peptide components.

The observation that the two LEAPS conjugates act in a somewhat independent manner may have added advantages for therapy since it implies activation of both overlapping and different arms of protective mechanisms. This offers advantages in case one vaccine is not effective in an individual. The mixture of DerG-PG70 (CEL-4000) and DerG-PG275Cit also provides immunotherapy for responses elicited by more than one epitope. Such a combination vaccine would offer advantages in cases when both or only one of the PG epitopes are recognized by immune cells within a patient [18].

In vivo promotion of Th2 responses in GIA mice by CEL-4000 was observed in our previous work [21] and also confirmed in the present study. The question arose whether the anti-inflammatory cytokine dominance, induced by the CEL-4000 vaccine, was the result of up-modulation of Th2 responses or down-modulation of Th1 and/or Th17 responses. The analysis of the effects of the CEL-4000 conjugate on cytokine production by T cells from unvaccinated GIA mice upon their restimulation with rhG1 antigen and in vitro differentiation into Th cell subsets clearly indicate that CEL-4000 promotes Th2 responses but does not down-modulate the production of signature cytokines by Th1 or Th17 cells. The inability of CEL-4000 to affect cytokine production by any subset of in vitro differentiated Th cells obtained from naïve BALB/c mice, despite polyclonal activation by anti-CD3 and anti-CD28 antibodies, strongly suggests that the CEL-4000 conjugate acts in an antigen (rhG1)-specific manner. However, the concentrations of signature cytokines secreted by each of the Th subsets following polyclonal activation of T cells from naïve mice were much higher than those induced in cells from GIA mice by rhG1 restimulation in vitro. Therefore, we cannot rule out the possibility that the lack of the in vitro effect of CEL-4000 on the polyclonally activated cells was due to the inability of this LEAPS peptide to override the excess of cytokine production mounted by the cells in response to the anti-CD3/CD28 antibodies.

Th2 responses to PG70 and DerG containing peptides (including, but not limited to, DerG-PG70/CEL-4000, DerG-PG275Cit and their combination) were all up-modulated in the rhG1-restimulated cultures of spleen CD4+ cells and APCs from unvaccinated GIA mice. Amplification of the Th2 response was not seen in the presence of the J-PG70 peptide. This is consistent with previous studies that have shown that DerG-LEAPS peptides elicit Th2 responses and J-LEAPS peptides elicit Th1 responses. This is also consistent with the inability of J-PG70 to elicit therapy in the GIA model [21] since promotion of Th1 responses could reinforce the Th1 pro-inflammatory responses that drive disease in mice with GIA.

## 5. Conclusions

By eliciting Th2 responses, vaccination with CEL-4000 and/or DerG-PG275Cit is shifting the immunopathological, inflammatory joint disease-driving Th1 response of the GIA model of RA, to a benign response to reset the immune balance in an antigen-specific manner. This is similar to the action of CEL-2000 [16], a J-LEAPS vaccine containing a peptide from human collagen type II, which elicits a Th1 response to immunomodulate Th17 driven disease in the CIA model of RA [26].

In summary, both CEL-4000 and DerG-PG275Cit LEAPS conjugates are therapeutic in the GIA model of RA with high efficacy, although they appear to act by different mechanisms. They work by enhancing Th2 and possibly by inducing Treg-like responses in this Th1 driven model. A combination vaccine containing both DerG LEAPS conjugates would offer advantages in case one epitope or another is not recognized by the disease-promoting T cell repertoire of a patient with RA. In contrast to current therapy, which treats symptoms or inhibits or ablates inflammatory mediators, CEL-4000 or the combination vaccine therapy provides an antigen specific immunomodulation of the disease driving immune responses to block disease progression.

## Figures and Tables

**Figure 1 vaccines-09-00448-f001:**
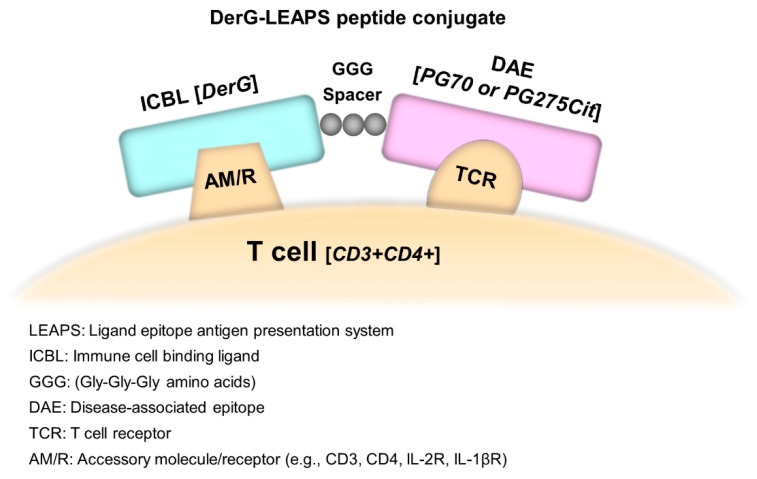
Model mechanism of action of a DerG-LEAPS conjugate. Based on findings of [21] and [3] for CEL-4000 (DerG-PG70) and earlier studies [29,30,31,32] for the G or DerG element. For DerG-LEAPS, the epitope peptide binds to the TCR of cognate CD4+ cells, while the DerG portion can bind to another receptor (including CD3, CD4, or other accessory molecules). A glycine triplet (GGG) is inserted as a spacer between the DerG and epitope peptides. DerG-LEAPS can promote the secretion of anti-inflammatory cytokines (IL-4, IL-10) by CD4+ Th2 or Treg cells. For J-LEAPS conjugates, the initial interaction of the J-ICBL promotes interaction with a DC which initiates maturation, IL-12p70 secretion, and antigen presentation to CD8+ T cells [17] (IL-1βR and IL-2R are receptors for IL-1β and IL-2, respectively).

**Figure 2 vaccines-09-00448-f002:**
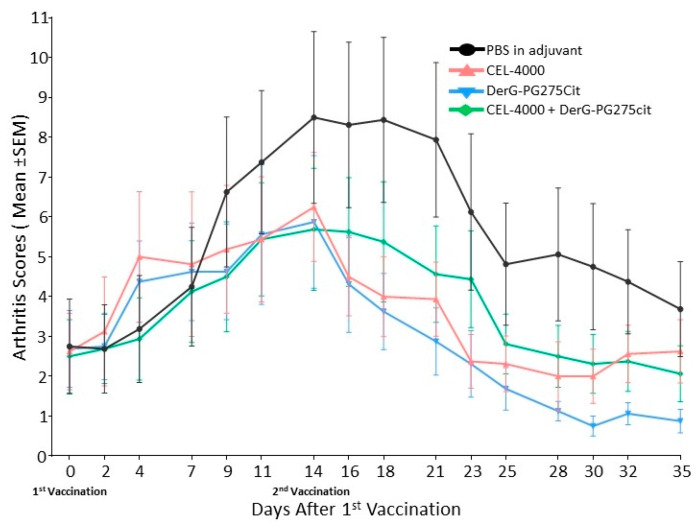
Therapeutic effects of LEAPS vaccine treatment on arthritis in GIA mice in the introductory study (study 1). LEAPS therapy was initiated in the GIA mice on days 0 and 14 after the onset of arthritis (visual arthritis score ~3 in each group, *n* = 8 mice per group). All treatments were administered in adjuvant. Arthritis severity was assessed by visual scoring 3 times a week for 5 weeks. The treatment groups were as follows: PBS in adjuvant (adjuvant control) represented in black; LEAPS conjugate DerG-PG70 (CEL-4000), represented in red; LEAPS conjugate DerG-PG275Cit, represented in blue; Combination of CEL-4000 and DerG-PG275Cit, represented in green. Disease severity was significantly suppressed (reduced arthritis score) in mice treated with LEAPS vaccines as compared with the adjuvant-treated control group. The 2-way repeated measures ANOVA was used for statistical analysis (see Appendix A for *p* values of visual arthritis scores and other details).

**Figure 3 vaccines-09-00448-f003:**
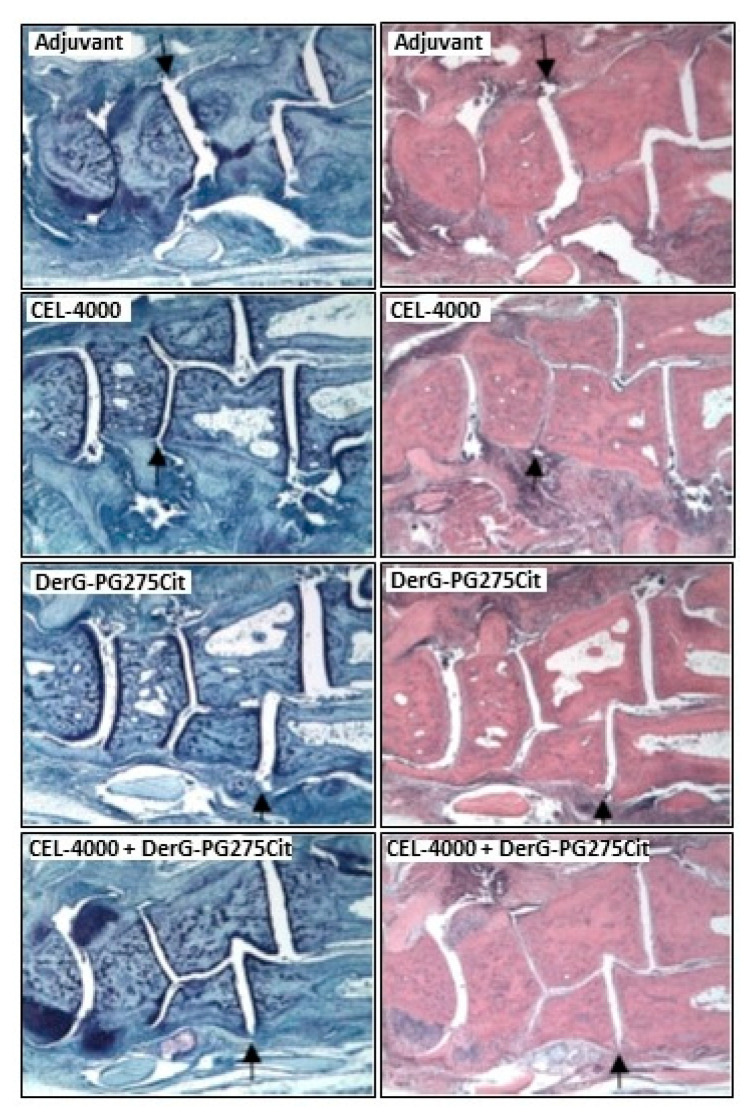
Representative histopathology images of ankles and foot joints of mice from the introductory study (Study 1). Adjacent tissue sections of ankles and midfoot joints from adjuvant-treated control or LEAPS vaccine treated GIA mice were stained with toluidine blue (TB, left-side images) or hematoxylin and eosin (H&E, right-side images). TB staining is well suited to joint sections, as it depicts matrix loss from damaged cartilage, and bone resorption (lighter than normal staining) as well as periosteal bone formation (darker than normal staining). TB-stained sections can also be assessed for inflammation, pannus formation and overall tissue damage. However, H&E staining captures inflammation (leukocyte infiltration) better than TB. The sections from the adjuvant-treated animal show marked inflammation, moderate cartilage damage, mild pannus and bone resorption as well as severe periosteal bone formation. The animal vaccinated with CEL-4000 displays moderate inflammation and mild cartilage damage in one joint only. The sections from a DerG-PG275Cit-treated animal show moderate inflammation and mild cartilage damage in several joints. The animal treated with the combination of CEL-4000 and DerG-PG275Cit demonstrates mild inflammation and cartilage damage, mild pannus and bone resorption as well as moderate periosteal bone formation in a single joint. A black arrow points to a pathological change in each of the joint sections. Inflammatory and structural damages were the most widespread in the joints of the adjuvant-treated control mouse. Original magnification: 40x.

**Figure 4 vaccines-09-00448-f004:**
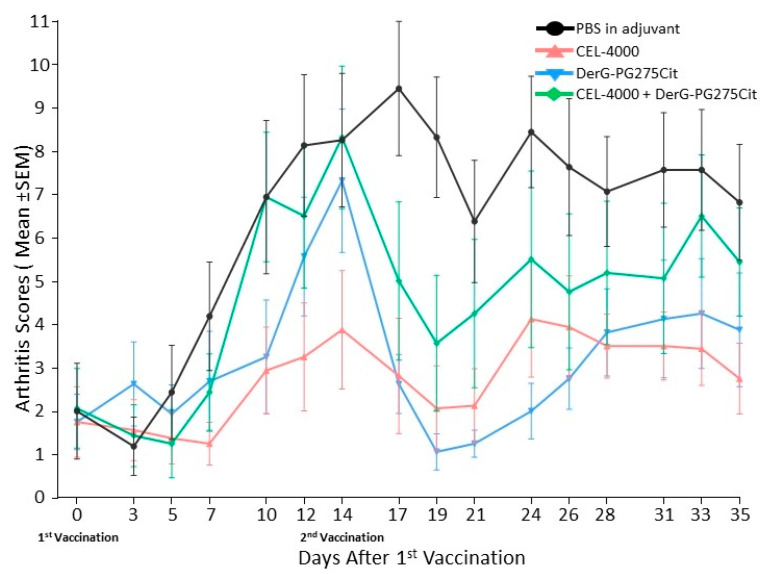
Therapeutic effects of LEAPS vaccine treatment on arthritis in GIA mice in the confirmatory study (study 2). LEAPS therapy was initiated in the GIA model after the onset of arthritis. The groups and treatment protocols for control (adjuvant) and LEAPS administration were the same as in Figure 2. Disease severity was significantly suppressed (as indicated by reduced arthritis score) in mice treated with LEAPS vaccines as compared with the adjuvant-treated control group (*n* = 8 mice per group). The 2-way repeated measures ANOVA was used for statistical analysis (see Appendix A for *p* values of visual arthritis scores and other details).

**Figure 5 vaccines-09-00448-f005:**
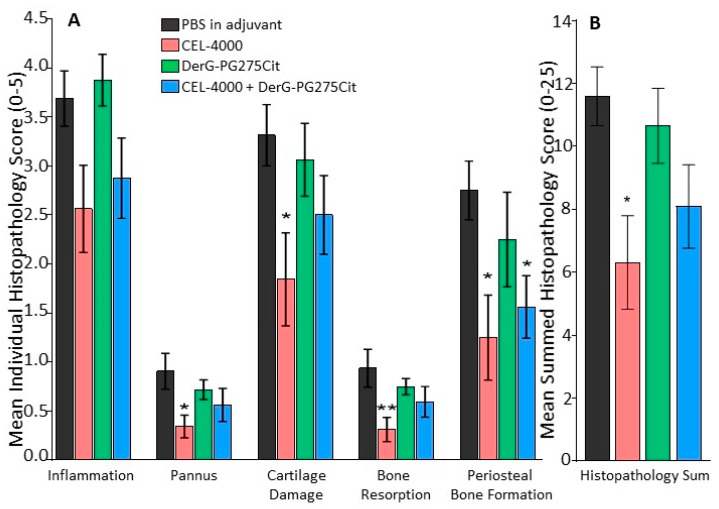
Analyses of histopathological data. Images of the joint sections of control and LEAPS treated GIA mice (from study 2) were examined and scored by an independent team of certified pathologists. TB- and H&E-stained sections of hind limbs from coded GIA mice (*n* = 8 per group) were photographed and scored for the degree of (**A**) inflammation, pannus formation, cartilage damage, bone resorption, and periosteal bone formation. Each of these parameters received a score ranging from 0 to 5. Hind limbs from naïve (non-immunized) female BALB/c mice were also processed for histology for reference to normal joint structure (scored as 0). (**B**) Summed histopathological scores were calculated from the above-listed individual scores. The results are in agreement with prior reports for CEL-4000-treated and control animals as well as with the in vivo findings shown in Figure 4. The non-parametric Kruskal–Wallis test, one-Way ANOVA, and Dunn’s multiple comparison test were used for statistical analysis (* *p* ≤ 0.05, ** *p* ≤ 0.01). Representative TB-stained joint section images from each group of control and LEAPS vaccinated GIA mice are shown in Appendix A.

**Figure 6 vaccines-09-00448-f006:**
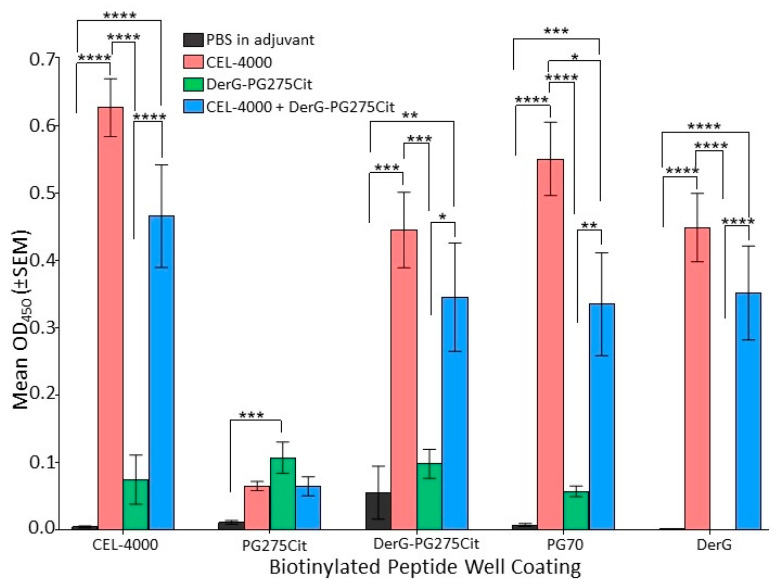
Serum anti-peptide antibody levels of vaccinated GIA mice. Anti-peptide serum antibodies of GIA mice from study 2 (*n* = 8 mice per group) were assayed at a dilution of 1:1000 using biotinylated peptide-coated Neutravidin plates by ELISA. The X axis lists the biotinylated peptides used to coat the Neutravidin-containing wells. Peptide-bound IgG antibodies were detected using anti-mouse IgG secondary antibody. Optical Density (OD) values at 450 nm are shown on the Y axis. OD450 readings of the OVA peptide (background control) were subtracted from each of the other peptide-specific readings. One-way ANOVA with Tukey’s multiple comparison test were used for statistical analysis (* *p* ≤ 0.05, ** *p* ≤ 0.01, *** *p* ≤ 0.001, **** *p* ≤ 0.0001).

**Figure 7 vaccines-09-00448-f007:**
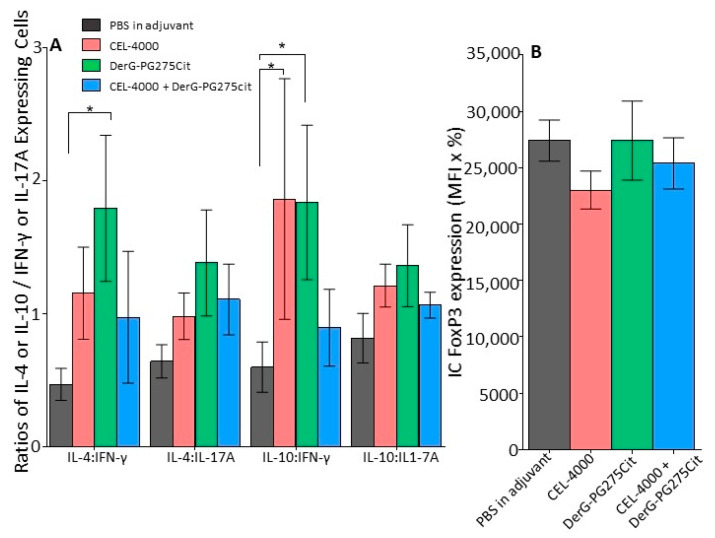
Intracellular (IC) cytokine and FoxP3 expression of spleen cells from control and LEAPS vaccine treated GIA mice. Spleen cells collected from the GIA mice at the end of study 2 were analyzed by flow cytometry for IC expression of cytokines or the Treg marker FoxP3. Expression data were calculated by multiplying mean fluorescence intensity (MFI) by the percent (%) of cytokine- or FoxP3 positive CD4+ T cells. (**A**) Ratios of anti-inflammatory (IL-4 or IL-10) to pro-inflammatory (IL-17A or IFN-γ) cytokines detected inside the GIA CD4+ cells. (**B**) IC FoxP3 expression in CD4+ spleen cells of each group of mice (MFI × %). Statistical analysis of the IC staining data was performed using (**A**) 2-way ANOVA with Fisher’s LSD test or (**B**) Kruskal–Wallis test with Dunn’s multiple comparison (*n* = 5 mice per group, * *p* ≤ 0.05).

**Figure 8 vaccines-09-00448-f008:**
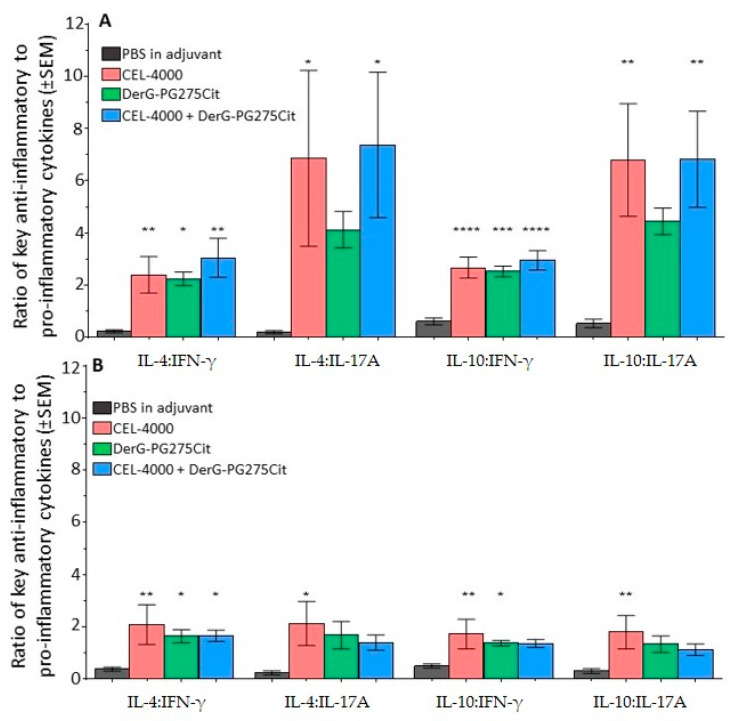
Cytokine production by spleen cells from control and LEAPS vaccine treated GIA mice. Spleen cells obtained from control or vaccinated GIA mice (*n* = 8 per group) from study 2 were either non-restimulated or restimulated with rhG1 antigen in culture for 4 days and then media samples were assayed for released cytokines. Cytokine results were obtained using Millipore’s Luminex MagPix instrument and R&D Systems’ Mouse Magnetic Luminex Assay kits (Biotechne brand). Results are presented as ratios of key anti-inflammatory to pro-inflammatory cytokines (to be comparable with the intracellular cytokine ratios shown in Figure 7) secreted by (**A**) Non-restimulated cell cultures and (**B**) Antigen-restimulated cell cultures (* *p* ≤ 0.05, ** *p* ≤ 0.01, *** *p* ≤ 0.001, **** *p* ≤ 0.0001). Statistical analysis was performed using one-way ANOVA and Fischer’s LSD multiple comparison tests. The concentrations of these four and a few additional cytokines are shown in Appendix A.

**Figure 9 vaccines-09-00448-f009:**
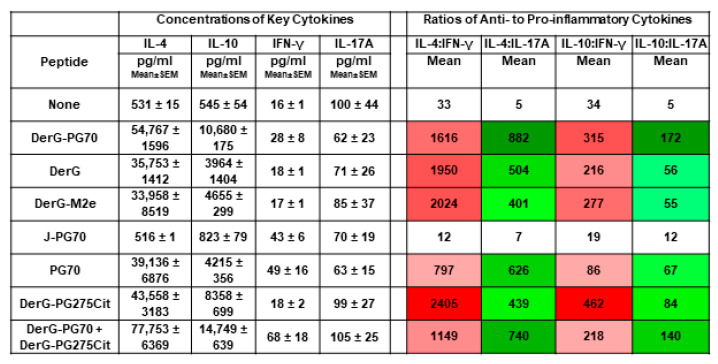
The effects of CEL-4000 and other peptides on the in vitro differentiation of Th2 cells from unvaccinated GIA mice. CD4+ spleen cells isolated from unvaccinated GIA mice were co-cultured with GIA spleen antigen-presenting cells, rhG1, and without (None) or with the indicated peptides in the presence of Th2 differentiation promoting reagents for a 6-day differentiation period, then the reagents were washed out, and the peptides and rhG1 were added back for a 2-day post-differentiation period. Results are expressed as concentrations.

## Data Availability

The data presented in this study will be filed and openly available in available in FigShare at https://doi.org/10.6084/m9.figshare.14396168.v1 (accessed on 9 April 2021).

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
