# Peer review of "Vaccination by Two DerG LEAPS Conjugates Incorporating Distinct Proteoglycan (PG, Aggrecan) Epitopes Provides Therapy by Different Immune Mechanisms in a Mouse Model of Rheumatoid Arthritis"

_vaccines, 2021, doi:10.3390/vaccines9050448_

Round 1

Reviewer 1 Report

This is a thorough study trying to offer therapy via vaccination to the devastating disease the Rheumatoid Arthritis and to explain the mechanism of action of the compounds used. Yet this study is impossible to read and to follow. Therefore in order to enable the reader to understand what is this study about the following should be considered:

  1. Schematic representation of LEAPS technology e.g. the one presented at Cel-Sci company website.
  2. Introduction of e.g. Compound A, Compound B, Compound C instead of the very long names and abbreviations.
  3. Real results (concentrations) of the various cytokines measured should be provided on top of the presented ratios.
  4. The ratios should be calculated in molarity since the M.W. of the various cytokines is not the same and it is not clear whether the delta in the concentrations cover up for it.
  5. Do not send the reader to a previous study to look for a main method (ratio) used in the present study.
  6. The novelty of the present study over the studies presented in Ref. 15 and Ref. 25 by the same group should be discussed.
  7. The therapy suggested should be discussed vis a vis existing RA drugs.
  8. In the introduction, references from 2007 and 2012 (rows 48 and 49) are provided to present RA in numbers. I believe the CDC had updated these numbers since.
  9. Row 48 and 50 erase the full stop after USA.

Author Response

Responses to Reviewer 1 Comments

General Comments:

This is a thorough study trying to offer therapy via vaccination to the devastating disease the Rheumatoid Arthritis and to explain the mechanism of action of the compounds used. Yet this study is impossible to read and to follow. Therefore, in order to enable the reader to understand what this study about the following should be considered:

Response to General Comments: We would like to thank this reviewer for the time and effort extended in reviewing our manuscript.  In our attempt to limit the length and verbiage of the manuscript, some of the aspects that this reviewer is suggesting were not included in the body of the manuscript but now included in either the main text, revised supplementary materials or the repository deposited materials.  These materials will be available to the reviewers. 

Specific comments:

Point 1: Schematic representation of LEAPS technology e.g. the one presented at Cel-Sci company website.

Response 1: We have prepared and inserted a DerG-LEAPS model with a brief legend as Figure 1(final version lines 101-107)  and thank the reviewer for suggesting the opportunity.  We laud the reviewer for looking for more information of the LEAPS technology and the current Figure 1 is a better indicator of the mechanism than the diagram that is presented on the CEL-SCI website, which is outdated and overly simplistic, and our earlier review articles present only graphic models focused mainly for the J-LEAPS vaccines.  

Point 2: Introduction of e.g. Compound A, Compound B, Compound C instead of the very long names and abbreviations.

Response 2: We respectfully disagree with the Reviewer that providing generic labels for the vaccines as compound A, B, C or D would be helpful. It would likely cause confusion. The vaccine names are descriptive of their component peptides and are appropriate.

Point 3: Real results (concentrations) of the various cytokines measured should be provided on top of the presented ratios.

Response 3: The concentrations of the cytokines for each set of data are presented in the main text (Fig 9, and in supplementary materials (Fig. S2,S3 and S4). Inclusion of these numbers in the main manuscript text would obscure the results of the experiment and unnecessarily complicate the figure and the message it is portraying.

Point 4: The ratios should be calculated in molarity since the M.W. of the various cytokines is not the same and it is not clear whether the delta in the concentrations cover up for it.

Response 4: The results are expressed as ratios to be consistent with previously reported findings in Mikecz et al 2017.[21] By comparing the cytokine data for the different vaccine treatments as ratios and only comparing the vaccine results within a cytokine ratio, it does not matter whether the data is based on the number of grams or the number of moles of the cytokines since ratios are unitless.  In addition,  each value would be multiplied by a common ratio based on the grams per mole for both factors which would give the same ratio.

Point 5: Do not send the reader to a previous study to look for a main method (ratio) used in the present study.

Response 5: Reference to published methods is a common approach in manuscripts and facilitates the reading of the manuscript. We felt when submitted the data analysis ratio approach was sufficiently described in this manuscript. The ratio method used in the present study is briefly described in the methods section 2.9 (Measurements of cytokines) [final version lines 266-269] where it is indicated that “The data shown in the figures was expressed as the concentrations of key secreted (IL4, IL10, IFNγ, and IL17) cytokines, or as the ratios of anti-inflammatory (IL4 or IL10) to pro-inflammatory (IFNγ or IL17A) cytokines, as described previously.” with a reference to our previous publication [21] for additional details.  However as requested, we added more details to the manuscript to further clarify the ratio method used in the present study (final version lines 270-271 ).

Point 6: The novelty of the present study over the studies presented in Ref. now 21 15 (Mikecz et al) and Ref. now 35 25 (Markovitz et al) by the same group should be discussed.

Response 6: The novelty of the present study is presented in more depth. We modified the discussion to point out these two major distinctions by adding the following points in this section. Mikecz et al 2017 [21] examined only one DerG conjugate using only a single disease associate epitiope (PG70).  In addition that study only examined spleen cells with or without antigen in vitro activation of antigen and no studies in either study of differentiation conditions from Th0 to Th1, Th2, Th17 or Treg cells or examination of the effect of the DerG conjugates or other peptides on the cytokines that were elicited from diseased GIA spleen cells. We further showed that educated (antigen-experienced) spleen cells  from diseased animals were essential and naïve spleen cells even if differentiated in vitro and activated polyclonally were not affected by the DerG-PG70 conjugates in vitro. Furthermore Markovics et al 2016 [35] did not examine any LEAPS conjugates either in vivo or in vitro on the progression of disease or spleen cells from GIA mice(final version lines 525-532).

Point 7: The therapy suggested should be discussed vis a vis existing RA drugs.

Response 7: The LEAPS therapy is contrasted to existing RA drugs in more depth in the introduction. It should be noted and it is mentioned that Rosenthal et al discussed other therapies for RA and in relation to LEAPS conjugates and the differences in mechanism of action and targets in an earlier article3 . We modified the manuscript in several  places to point this out(final version line 62-67 and 610-613 ).

The first paragraph of the introduction states, “RA biologics that treated arthritis symptoms by ablation of activated immune cells or neutralization of pro-inflammatory cytokines.” and then proceeds to introduce the LEAPS vaccines as promoting immunomodulation, an alternative approach.  According to this reviewer’s suggestion, we added several articles (references 8-15)  and the following sentence to the conclusion: (final version lines 610-613) “In contrast to current therapy, which treats symptoms or inhibits or ablates inflammatory mediators, CEL-4000 or the combination vaccine therapy provides an antigen specific immunomodulation of the disease driving immune responses to block disease progression.” .

Point 8: In the introduction, references from 2007 and 2012 (rows 48 and 49) are provided to present RA in numbers. I believe the CDC had updated these numbers since.

Response 8: We have provided a more recent (2019) information about the incidence of RA (reference 2, cited in final version lines 49-50)

Point 9: Row 48 and 50 erase the full stop after USA.

Response 9: The errors have been corrected(final version line 48 ).

Reviewer 2 Report

The ms of Zimmerman et al describes two small animal studies of peptide vaccines in the GIA model of rheumatoid arthritis. The two vaccines studied, administered separately or alone, are claimed to induce therapeutic effects in this model, despite somewhat divergent humoral and cellular responses. Although the data appear robust and may be of some interest to readers of Vaccines, several aspects of interpretation appear to be difficult to support on the strength of the data as it is currently presented.

The claimed therapeutic effect of DerG-PG275Cit is evident only in gross morphological scores, with no evidence of improvement in histopathological readouts of disease. As such, the interpretation that "the two peptides elicit different therapeutic immune responses" (Abstract, line 33, and similar claims in the discussion) seems difficult to sustain in teh face of the alternative that the DerG-PG275Cit response is simply inferior to the CEL-4000 response.

Likewise the subsequent claims that "the combination of the two LEAPS conjugates may provide broader epitope coverage and greater efficacy" is entirely unsupported by data. Epitope coverage is not assessed at all in the ms, and Figs 1, 3 and 4 show the combination to be, at best, equal to CEL-4000 alone.

Given most readers of Vaccines are not RA experts, further guidance to the interpretation of Fig 2 would be helpful. The differences between control and vaccinated images are far from obvious to the untrained eye.

Figs 1 & 3 present a continuous variable (time) on a categorical axis (the gap between day 2 and day 4 should be smaller than the gap between day 4 and day 7

Author Response

Responses to Reviewer 2 Comments

General Comments:

The ms of Zimmerman et al describes two small animal studies of peptide vaccines in the GIA model of rheumatoid arthritis. The two vaccines studied, administered separately or alone, are claimed to induce therapeutic effects in this model, despite somewhat divergent humoral and cellular responses. Although the data appear robust and may be of some interest to readers of Vaccines, several aspects of interpretation appear to be difficult to support on the strength of the data as it is currently presented.

Response to General Comments: We would like to thank this reviewer for the time and effort extended in reviewing our manuscript. We provide a response to each point below:

Point 1: The claimed therapeutic effect of DerG-PG275Cit is evident only in gross morphological scores, with no evidence of improvement in histopathological readouts of disease. As such, the interpretation that "the two peptides elicit different therapeutic immune responses" (Abstract, line 33, and similar claims in the discussion) seems difficult to sustain in teh face of the alternative that the DerG-PG275Cit response is simply inferior to the CEL-4000 response.

Response 1:   The data for the DerG-PG275Cit in figures 2-5 demonstrate a clear therapeutic benefit from treatment that can be distinguished from the adjuvant only treated mice from day 17 in both

studies. Unlike CEL-4000, no significant antibody was elicited by DerG-PG275Cit  although there was a favorable anti-inflammatory cytokine response. This shows a difference in immunological action for the two peptides even though both are effective. It is significant that both peptides have immunomodulatory activity on the cytokine ratios and more importantly both peptides independently have therapeutic activity. Superiority or inferiority is difficult to assign. More importantly, the take home lesson is that a combination is a viable treatment that has potential advantages.

Point 2: Likewise the subsequent claims that "the combination of the two LEAPS conjugates may provide broader epitope coverage and greater efficacy" is entirely unsupported by data. Epitope coverage is not assessed at all in the ms, and Figs 1, 3 and 4 show the combination to be, at best, equal to CEL-4000 alone.

Response 2: By administering two peptides, both of which have independent  activities, each containing different epitopes, as identified by other studies, validates the statement that more than one (meaning broader) epitope coverage is provided with the potential, in some cases, for greater efficacy and T cell antigenic coverage. Equal efficacy to CEL-4000 is an examplary level of activity. We added “in some cases” to the Abstract (final version line 35).

Point 3: Given most readers of Vaccines are not RA experts, further guidance to the interpretation of Fig 2 would be helpful. The differences between control and vaccinated images are far from obvious to the untrained eye.

Response 3: We looked at in-life as well as histological and cytokine profile findings. We provided additional details to the description of histopathological changes in Figure 3(final version line 339-357 )  and supplemental Figure S1(Fig S1 legend).

Point 4: Figs 1 & 3 present a continuous variable (time) on a categorical axis (the gap between day 2 and day 4 should be smaller than the gap between day 4 and day 7

Response 4: The graphs have been corrected(final version lines 318-319 ) and (lines 365-366 ).

Reviewer 3 Report

Authors have designed a study to evaluate therapeutic effects of cartilage proteoglycan epitope derived conjugate peptide vaccines DerG-PG70, DerG-PG275Cit, or combination of both peptides towards Rheumatoid arthritis by examining mouse model of RA.   

Strong point of this manuscript is the reduced arthritis progression and mild inflammation and cartilage damage caused possibly by upregulation of Th2 cell resulting in high anti-inflammatory cytokine expression.

I find this manuscript well conceptualized and very well written. Results are clearly presented, and previous literature reports are well cited. This manuscript would interest readers and provide valuable information in Rheumatoid arthritis research. However, following minor concerns should be addressed.

  1. Although the outcome of the both study 1 and 2 are the same (i.e. reduction of arthritis progression), the arthritis scores for the control and other vaccinations are significantly different. For example, arthritis score for DerG-PG275Cit is 1 in study-1 vs 4 in study-2. Can authors offer any explanation for this significant difference?
  2. It’s clearly visible from Fig. 7A that concentration of IL4 and IL10 is high compared to IFNγ to provide higher anti/pro-inflammatory cytokines ratio. However, in the case of IL4-IL10/IL17A, the ratio is too small. Is it because of the upregulation of Th17 cells in this case? If it is the case, then please offer an explanation in the manuscript text.
  3. Line 452-545, it’s said that CEL-4000 is the most effective anti-inflammatory cytokines inducer, but DerG-PG275Cit appears to induce anti-inflammatory cytokines in the same magnitude, atleast in case of IL4-IL10/IFNγ. Should both CEL-4000 and DerG-PG275Cit be mentioned as the most effective inducers?

Author Response

Responses to Reviewer 3 Comments

General Comments:

Authors have designed a study to evaluate therapeutic effects of cartilage proteoglycan epitope derived conjugate peptide vaccines DerG-PG70, DerG-PG275Cit, or combination of both peptides towards Rheumatoid arthritis by examining mouse model of RA.   

Strong point of this manuscript is the reduced arthritis progression and mild inflammation and cartilage damage caused possibly by upregulation of Th2 cell resulting in high anti-inflammatory cytokine expression.

I find this manuscript well conceptualized and very well written. Results are clearly presented, and previous literature reports are well cited. This manuscript would interest readers and provide valuable information in Rheumatoid arthritis research. However, following minor concerns should be addressed.

Response to General Comments: We would like to thank this reviewer for the time and effort extended in reviewing our manuscript.  We provide a response to each point below:

Point 1: Although the outcome of the both study 1 and 2 are the same (i.e. reduction of arthritis progression), the arthritis scores for the control and other vaccinations are significantly different. For example, arthritis score for DerG-PG275Cit is 1 in study-1 vs 4 in study-2. Can authors offer any explanation for this significant difference?

Response 1: Since the two in vivo treatment studies were performed at different times, and although treatment was initiated at similar aggregate disease levels, the individual mice had different presentations than the first trial, both of which can result in differences in overall treatment efficacy and disease time course. The variation could also have been due to seasonal or environmental factors. In our experience, variation is much less if two replicate (but independent) experiments are conducted simultaneously, which could not be done for this particular study. We do not have a solid explanation for this phenomemnon, but it is also observed in other arthritis models. We believe that the most important point is that the animals reacted similarly to LEAPS vaccine treatments in both experiments. We acknowledged the differences in disease course (final version lines 361-364 and line 374-375), but cannot offer an explanation for the reasons discussed above for seasonal and/or environmental factors.

Point 2: It’s clearly visible from Fig. 7A that concentration of IL4 and IL10 is high compared to IFNγ to provide higher anti/pro-inflammatory cytokines ratio. However, in the case of IL4-IL10/IL17A, the ratio is too small. Is it because of the upregulation of Th17 cells in this case? If it is the case, then please offer an explanation in the manuscript text.

Response 2: We updated and revised these figures based on audit of our studies and the comments of the reviewer.  We expanded the figures to include all cytokines assayed including the lower amounts of IL1b, IL2, IL6 and TNFa (Supplementary figure S2) not just the 4 selected and related to anti and pro inflammatory activity.  We frequently found that serum IL17 levels are low in mice with GIA, but their spleen cells secreted relatively high amounts of IL17 in culture. We can only speculate that IL17 might be in tissue or bound to receptors in vivo, but is free and accessible under in vitro culture conditions.

Point 3: Line 452-545, it’s said that CEL-4000 is the most effective anti-inflammatory cytokines inducer, but DerG-PG275Cit appears to induce anti-inflammatory cytokines in the same magnitude, atleast in case of IL4-IL10/IFNγ. Should both CEL-4000 and DerG-PG275Cit be mentioned as the most effective inducers?

Response 3: The text within this part of the manuscript was edited and now reads: (final version lines 452-455) “As shown in Fig. 8A, spleen cells from LEAPS vaccinated GIA mice all produced higher ratios of anti-inflammatory to pro-inflammatory cytokines than the cells from adjuvant-treated control animals, especially with regard to the IL4 or IL10 to IFNg ratios.” 

This is reiterated in final version lines 501-504: “As seen in Fig. 9 for Th2 differentiated GIA cells, the DerG and PG70 containing peptides elicited elevated ratios of anti-inflammatory to pro-inflammatory cytokines with CEL-4000 and DerG-PG275Cit being the most effective inducers of the anti-inflammatory cytokine response.”

Round 2

Reviewer 1 Report

1. The authors responded to and accepted most of the comments raised by the reviewer.

2. Hyphen should be added to all cytokines mentioned in the MS in the form of IL- (e.g. IL-6, IL-17 etc.) and not IL6 etc.

3. In row 531 add Markovics et al. (35).

4. Remove full stop were not meant e.g. in row 67.

5. Add space between words where needed.
